# PeerJ

# Removal of corallivorous snails as a proactive tool for the conservation of acroporid corals

Dana E. Williams[1,2], Margaret W. Miller[2], Allan J. Bright[1,2] and Caitlin M. Cameron[1,2]

[1] Cooperative Institute for Marine and Atmospheric Studies, Rosenstiel School of Marine and Atmospheric Science, University of Miami, Miami, FL, USA
[2] Southeast Fisheries Science Center, NOAA-National Marine Fisheries Service, Miami, FL, USA

Corresponding author
Dana E. Williams,
dana.williams@noaa.gov

## ABSTRACT

Corallivorous snail feeding is a common source of tissue loss for the threatened coral, *Acropora palmata*, accounting for roughly one-quarter of tissue loss in monitored study plots over seven years. In contrast with larger threats such as bleaching, disease, or storms, corallivory by *Coralliophila abbreviata* is one of the few direct sources of partial mortality that may be locally managed. We conducted a field experiment to explore the effectiveness and feasibility of snail removal. Long-term monitoring plots on six reefs in the upper Florida Keys were assigned to one of three removal treatments: (1) removal from *A. palmata* only, (2) removal from all host coral species, or (3) no-removal controls. During the initial removal in June 2011, 436 snails were removed from twelve 150 m$^2$ plots. Snails were removed three additional times during a seven month "removal phase", then counted at five surveys over the next 19 months to track recolonization. At the conclusion, snails were collected, measured and sexed. Before-After-Control-Impact analysis revealed that both snail abundance and feeding scar prevalence were reduced in removal treatments compared to the control, but there was no difference between removal treatments. Recolonization by snails to baseline abundance is estimated to be 3.7 years and did not differ between removal treatments. Recolonization rate was significantly correlated with baseline snail abundance. Maximum snail size decreased from 47.0 mm to 34.6 mm in the removal treatments. The effort required to remove snails from *A. palmata* was 30 diver minutes per 150 m$^2$ plot, compared with 51 min to remove snails from all host corals. Since there was no additional benefit observed with removing snails from all host species, removals can be more efficiently focused on only *A. palmata* colonies and in areas where *C. abbreviata* abundance is high, to effectively conserve *A. palmata* in targeted areas.

## INTRODUCTION

Predator control as a management strategy is most commonly considered for invasive predators (*Barbour et al., 2011*; *Baxter et al., 2008*; *Morris Jr, Shertzer & Rice, 2011*) or outbreaks of endemic predators (*Sanz-Aguilar et al., 2009*; *Yamaguchi, 1986*). Previous

attempts to cull corallivores, specifically *Acanthaster planci*, have largely been aimed at localized outbreaks with the goal of preserving coral tissue over a large area (*Yamaguchi, 1986*). These efforts have been deemed ineffective due to the large numbers and migrating aggregations of these predators (*Johnson, Moran & Driml, 1990*; *Yamaguchi, 1986*). However, removal of a relatively sedentary predator from targeted populations of a threatened coral species has not been evaluated.

Ecological theory on predator–prey dynamics can provide insight on situations when predator removal may be effective in protecting prey. *Sinclair et al. (1998)* present a framework whereby controlling natural predators may improve the outcome for management of declining or reintroduced populations of threatened species. In this framework, the appropriate scale of intervention depends on the functional and numerical response of predators to changing prey abundance. In cases where the effects of predation are depensatory and prey abundance is so low that they are vulnerable to stochastic events, predator control could be beneficial to prey populations (*Sinclair et al., 1998*). *Rotjan & Lewis (2008)*, in a review of corallivory, suggest that the rapid pace of coral decline over the past two decades, largely from factors other than predation, may have indeed reached a depensatory threshold such that predation is exerting undue influence and potentially compromising coral reef resilience.

On reefs in the western Atlantic, the dominant framework builder, *Acropora palmata*, is preyed upon by the corallivorous snail, *Coralliophila abbreviata*. Although disease, storms and bleaching have largely driven the range-wide decline of *A. palmata* populations, snail predation is recognized as one of the top three proximal threats to the persistence and recovery of *A. palmata* populations (*Bruckner, 2002*; *Williams & Miller, 2012*). In the upper Florida Keys, there has been a 50% decline in *A. palmata* tissue abundance since 2004. Although the main culprit has been disease, feeding by *C. abbreviata* accounted for an estimated one-quarter of the observed *A. palmata* tissue loss (*Williams & Miller, 2012*). As *A. palmata* populations decline, snails have been observed to become more concentrated on the remaining *A. palmata* (*Baums, Miller & Szmant, 2003a*; *Bruckner, 2000*; *Bruckner, Bruckner & Williams, 1997*; *Williams & Miller, 2012*) rather than declining themselves, suggesting increasing per capita impact on prey.

*Coralliophila abbreviata* preys on multiple coral host species including acroporids, *Orbicella* spp., *Diploria* spp., *Colpophylia natans*, *Agaricia* spp. and occasionally other mounding coral species (*Miller, 1981*). Snails found on *A. palmata* are larger, older, have higher fecundity (*Johnston & Miller, 2007*) and consume more coral tissue than on other coral host species (*Bruckner, 2000*). These snails are typically found in groups (*Baums, Miller & Szmant, 2003a*; *Bruckner, 2000*; *Bruckner, Bruckner & Williams, 1997*) feeding on coral tissue, leaving behind a feeding scar of exposed skeleton. They are relatively sedentary, often remaining on a prey colony until no living tissue remains, at which point they migrate to a neighboring colony (*Bruckner, 2000*; *Williams & Miller, 2012*). Individual snails can consume up to 16 cm$^2$ of tissue per day (*Baums, Miller & Szmant, 2003b*; *Brawley & Adey, 1982*), though they do not feed continuously throughout the year at that rate (*Bruckner, Bruckner & Williams, 1997*). In addition to directly removing *A. palmata* tissue

during feeding, *C. abbreviata* may indirectly affect corals by way of vectoring disease (*Williams & Miller, 2005*) or attracting other predators such as butterflyfish (*Brawley & Adey, 1982*) and *Hermodice carunculata* (DE Williams, pers. obs., 2006). Thus, *C. abbreviata* has substantial direct and potential indirect effects on *A. palmata*. Because this predator has low mobility and a relatively long lifespan (up to 15 years; *Johnston & Miller, 2007*), it may be feasible to locally reduce their abundance to conserve *A. palmata* tissue.

Acropora palmata was listed as threatened under the US Endangered Species Act (*NMFS, 2006*; *NMFS, 2012*) based on devastating declines throughout its range. The ESA listing carries with it a mandate to pursue management actions to foster recovery of the species (*US Endangered Species Act, 2013*, section 4f). Although predation is not the primary factor causing decline of this species, recent trajectories suggest it may be a fundamental factor inhibiting recovery and, at present, predation may be the most locally tractable threat. Even in regions where *A. palmata* is relatively rare, such as the Florida Keys, its distribution is clumped making targeted removal efforts logistically feasible. Therefore, ecological, legal and management conditions all point to the removal of *C. abbreviata* as a potential conservation action that could be feasible at the local level. Earlier work (*Miller, 2001*) showed that removing *C. abbreviata* snails on a colony scale can conserve *A. palmata* tissue, but nothing is known about the effectiveness in terms of recolonization rates or on a larger 'reef scale'. The current study utilized long-term fixed monitoring plots of *A. palmata* colonies to conduct experimental *C. abbreviata* (here after 'snail') removals to (1) determine the rate at which snails recolonize *A. palmata* colonies, (2) evaluate detectable impacts on the host *A. palmata* population, (3) compare the size distribution of recolonizing versus original snail populations and (4) evaluate the effort required for removal by resource managers.

## METHODS

Long-term *Acropora palmata* demographic monitoring plots (7 m radius) at six shallow sites in the upper Florida Keys National Marine Sanctuary (FKNMS) reef tract were used to implement a Before-After-Control-Impact (BACI) type design (*Green, 1979*; *Smith, 2006*) to evaluate the effects of snail removal. This design is useful in natural settings because initial variation among individual plots can be partitioned from treatment effects by comparing each plot's trajectory over time (before vs. after a manipulation) among plots subjected to different treatments. Each site included three plots which were haphazardly assigned a designation of one to three when initially established for monitoring (one to six years prior to the start of the experiment). Three snail removal treatments were assigned according to plot number: plots numbered '1' had snails removed from *A. palmata* colonies only ("Ap Only"); plots numbered '2' had snails removed from all host corals in the plot ("All Hosts"; mainly *A. palmata, Diploria* spp., *Orbicella* spp. and *Colpophylia natans*), and plots numbered '3' were designated "Control" in which snails were counted on the host corals in the plots, but were not removed. Thus, the assignment of plots to treatments, though not fully random, was based on a haphazardly-assigned numbering scheme applied years before with no anticipation of conducting the present experiment.

## Removal

This experiment proceeded in two phases. The first, 'removal phase' evaluated short-term reinfestation of the *A. palmata* colonies, and consisted of three removals beginning with an 'initial removal' of snails that provided the 'baseline' snail abundance and ending with a 'final removal'. Three removals were implemented during this phase according to logistic considerations. The second, 'recolonization phase' evaluated the 'effect duration' which is the projected time to reach the baseline snail abundance following a one-time removal effort. This 'recolonization phase' consisted of five surveys beginning after 'final removal' and ending at the 'final survey' when the snails were collected for analysis.

Removal of snails from host corals was performed by two SCUBA divers that were experienced (>5 years) in finding this somewhat cryptic gastropod species. Search procedures included tactile and visual examination of colony margins and undersides as *C. abbreviata* are typically encrusted by algae and therefore not visually apparent. Individual host colonies were searched, and when snails were found, the diver recorded the host species and number of snails on the colony. In the removal plots, snails were then removed and placed in separate zip-top bags according to host species. Snail shell length was measured to the nearest 0.1 mm using Vernier calipers. Due to the high abundance and small size of *Agaricia* spp. colonies, it was not feasible to systematically locate all the colonies. However, when *Agaricia* colonies were encountered in the All Hosts and Control plots, they were searched, and snails found in the All Host plots were removed. Initial removal was conducted during 14–16 June 2011 and the number of snails found was considered the 'baseline' snail abundance for each plot.

Snails were removed from all *A. palmata* colonies in the treatment plots again in early July 2011, September 2011 and January 2012 ('removal phase'). Thereafter, snails found on *A. palmata* were counted but not removed in May 2012, September 2012, January 2013, May 2013 and August 2013 ('recolonization phase'). At the final survey in August 2013, all host colonies were measured (length, width, height and % live) and surveyed for snails in all plots. Snails were collected from all host corals in the All Hosts treatment plots and *A. palmata* only in the Ap Only treatment plots and measured to compare the recolonized population with the initial population. Shells were crushed to determine the presence (designating males) or absence (designating females) of a penis. The time spent searching the host colonies and removing encountered snails was recorded during the final survey and the averages among plots were used to evaluate the 'effort' of each removal treatment.

During the removal and recolonization phases, routine monitoring of *A. palmata* in the study plots continued as described in *Williams, Miller & Kramer (2008)*. Once per year (fall), all *A. palmata* colonies in each plot were counted and the length, width and height were measured (using a meter stick), and % live tissue was visually estimated by a single observer (see *Williams, Miller & Kramer, 2006*; *Williams, Miller & Kramer, 2008*, for detailed survey protocols). Tissue abundance was estimated as a live area index (LAI), calculated by taking the colony's average dimension (average of length, width and height) squared and multiplying it by the visual estimate of percent live tissue cover. The LAI is summed for all colonies to get total tissue abundance for the 150 m$^2$ study plot.

Three times per year a randomly selected subset of tagged *A. palmata* colonies was further assessed for size, percent live tissue and presence of disease, snails and snail feeding scars. This work was conducted under permit numbers FKNMS-2010-033, FKNMS-2010-130 and FKNMS-2012-030 from the Florida Keys National Marine Sanctuary.

## Analyses

### *Effectiveness of removal*

We examined the total number of snails found on *A. palmata* colonies in the plot (ApSnails), the tissue abundance (LAI) of all *A. palmata* in the study plot (ApLAI) and the prevalence of disease and feeding scars among a subset of tagged *A. palmata* colonies. We used a BACI (Before-After-Control-Impact) design to compare the statistical interaction between time and treatment factors for the removal treatments and control. The total number of ApSnails and the ApLAI was compared among treatments using the initial (June 2011) and final (August 2013) surveys. Because the prevalence of both disease and feeding scars vary temporally (*Williams & Miller, 2012*), we compared the peak in prevalence from the year before the removal and the subsequent peak observed after the removal. Thus, disease prevalence was compared using fall 2010 (before) and fall 2011 (after) data and the prevalence of feeding scars were compared using spring 2011 (survey prior to removal) and spring 2012 data (first survey after final removal). Each parameter (ApSnails, ApLAI and prevalence of disease and feeding scars) was rank transformed to meet the assumptions of normality and homoscedasticity, and a repeated measures ANOVA was run on the ranks to look for significant ($p \leq 0.05$) within-subject interactions between time (the Before/After factor) and treatments (the Control/Impact factor) indicating that the trend in the measured parameter varied significantly between treatments.

### *Recolonization*

We examined the rate at which snails recolonized *A. palmata* colonies in removal plots over the recolonization phase. The number of snails present on *A. palmata* colonies following the final removal in January 2012 was assumed to be zero and the number found at each subsequent survey during this recolonization phase was plotted over time for each study plot and linear regression was used to determine the equation for the line. With "*y*" set to the baseline snail abundance for that plot, we solved for the projected date (*x*) that the number of snails found on the *A. palmata* colonies in that plot would return to its baseline abundance observed at the initial removal. The difference between the projected date and the date of the January 2012 removal was calculated as the treatment 'effect duration' for that plot. A Wilcoxon Matched-Pairs test was used to compare the baseline snail abundance, recolonization rate and effect duration between Ap Only and All Hosts removal treatments paired within site.

### *Snail size*

In order to compare the population of recolonized versus initial snails, shell length of the collected snails was measured and the data were log transformed to achieve normality. A two-way ANOVA was used to compare shell lengths between the two removal treatments

Table 1  Summary of coral host colonies and *Coralliophila abbreviata* snails found in the study plots at the initial and final survey. Observations at each study plot in three experimental treatments: removal of *Coralliophila abbreviata* snails from *Acropora palmata* only (Ap Only), from all coral host species (All Hosts) and controls in which snails were counted but not removed. Coral colonies and snails counted at the initial survey in June 2011 and the final survey in August 2013. *A. palmata* LAI (live area index) is calculated based on colony measurements as described in the text.

| Treatment | Reef | *A. palmata* colonies | | *A. palmata* snails | | *A. palmata* LAI (m$^2$) | | Other host colonies | | Other host snails | |
|---|---|---|---|---|---|---|---|---|---|---|---|
| | | Initial | Final | Initial | Final | Initial | Final | Initial | Final | Initial | Final |
| Ap Only | Carysfort | 32 | 21 | 43 | 16 | 8.4 | 8.4 | | 6 | | 4 |
| | Elbow | 55 | 76 | 27 | 19 | 15.2 | 14.8 | | 1 | | 0 |
| | French | 42 | 57 | 13 | 7 | 9.7 | 10.5 | | 19 | | 29 |
| | Key Largo Dry Rocks | 11 | 11 | 4 | 3 | 1.6 | 0.7 | | 5 | | 9 |
| | Molasses | 11 | 15 | 13 | 20 | 2.5 | 2.3 | | 0 | | 0 |
| | Sand Island | 18 | 21 | 32 | 17 | 2.8 | 4.7 | | 1 | | 1 |
| All Hosts | Carysfort | 8 | 13 | 17 | 6 | 6.7 | 2.3 | 3 | 3 | 61 | 34 |
| | Elbow | 28 | 31 | 20 | 9 | 5.0 | 4.8 | 2 | 1 | 10 | 3 |
| | French | 29 | 44 | 10 | 6 | 1.4 | 2.1 | 6 | 5 | 39 | 21 |
| | Key Largo Dry Rocks | 22 | 15 | 2 | 1 | 0.5 | 0.2 | 2 | 2 | 19 | 0 |
| | Molasses | 29 | 40 | 38 | 9 | 3.6 | 4.0 | 7 | 8 | 19 | 25 |
| | Sand Island | 41 | 51 | 60 | 39 | 12.7 | 16.7 | 1 | 1 | 9 | 4 |
| Control | Carysfort | 13 | 9 | 16 | 15 | 0.9 | 0.1 | 6 | 4 | 23 | 4 |
| | Elbow | 22 | 29 | 16 | 18 | 4.2 | 3.5 | 4 | 4 | 14 | 6 |
| | French | 24 | 20 | 8 | 5 | 1.9 | 1.2 | 15 | 15 | 29 | 31 |
| | Key Largo Dry Rocks | 10 | 9 | 11 | 10 | 3.0 | 2.5 | 6 | 6 | 10 | 10 |
| | Molasses | 22 | 31 | 15 | 21 | 8.7 | 8.7 | 7 | 6 | 7 | 8 |
| | Sand Island | 10 | 9 | 4 | 5 | 0.8 | 1.2 | 6 | 4 | 10 | 8 |

and time (initial removal vs. final survey). The size-frequency distributions of males and females at the final survey could not be compared between treatments due to small sample size ($n \leq 39$; Table 1), but the proportion of the population that was male was calculated as the number of male snails divided by the total number of snails.

## RESULTS

### Removal

Searching the host colonies in each 150 m$^2$ study plot required on average 30 diver minutes ($\pm 16$ min SD) at each removal for *A. palmata* only and an additional 21 diver minutes ($\pm 10$ min SD) when the other host species were searched. A total of 279 snails were removed from *A. palmata* in the twelve 150 m$^2$ removal treatment plots (Table 1), and a total of 157 snails were removed from other hosts in the 'all hosts' treatment plots. The baseline number of snails found on *A. palmata* in the control plots was notably (though not significantly) lower than in the removal treatment plots. This may be a result of a difference in detection as snails were counted in-place rather than removed in the controls, or it may be attributed to less live *A. palmata* tissue in the control plots (LAI; Table 1). Regardless of the cause for this difference, the BACI analysis is designed to reduce the effect of any initial differences in plots by evaluating only the change within each plot.

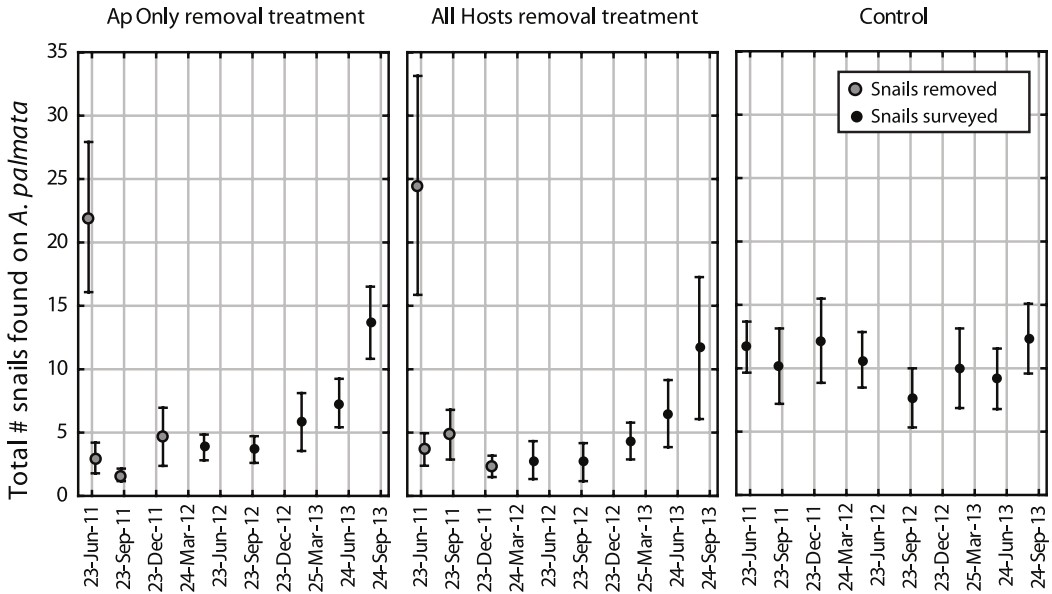

**Figure 1 _Coralliophila abbreviata_ abundance in experimental plots.** Number of _Coralliophila abbreviata_ found on _Acropora palmata_ per plot (mean ± SE). The initial removal occurred in June 2011, remaining snails were removed through January 2012 (removal phase, gray dots) after which they were only counted and left in place during the survey phase (solid black dots).

The mean number of snails found on _A. palmata_ colonies (ApSnails) in the removal plots remained less than five per plot during the removal phase, then gradually increased after removals stopped in January 2012 (Fig. 1). The interaction between time and treatment was significant for both the total number of ApSnails per plot (Fig. 2A; $p = 0.042$) and the prevalence of feeding scars (Fig. 2D; $p = 0.004$); for both, the removal treatments declined significantly while the controls remained unchanged. There was no change in tissue abundance (LAI) among all treatments (Fig. 2B). The prevalence of disease was significantly higher in fall 2011 compared to fall 2010 (Fig. 2C; $p = 0.016$) with no significant treatment effects or interaction.

## Recolonization

Despite high variability among individual removal plots, linear regressions of the number of snails found at each survey (Fig. 3) yielded $r$ values $\geq 0.7$ within each plot. Both the baseline abundance of ApSnails and the rate of snail recolonization (regression slopes of 0.002–0.050 snails d$^{-1}$) varied by an order of magnitude among reefs (Fig. 3). When treatment plots were paired by site, neither the baseline abundance of ApSnails nor the recolonization rate differed between treatments. If the Sand Island pair is excluded as an outlier (based on the extreme number of snails found in the Sand Island All Hosts plot, Table 1), then the baseline number of snails still does not differ between treatments, but the difference in recolonization rate between treatments becomes marginally significant (Wilcoxon, $Z = 2.02$, $n = 5$, $p = 0.043$). The baseline snail abundance and recolonization rate were highly correlated across all removal plots (Fig. 4). Overall, calculated 'effect durations' ranged from 1.3 to 5.9 years (Fig. 3) with an overall average of 3.7 ± 1.4

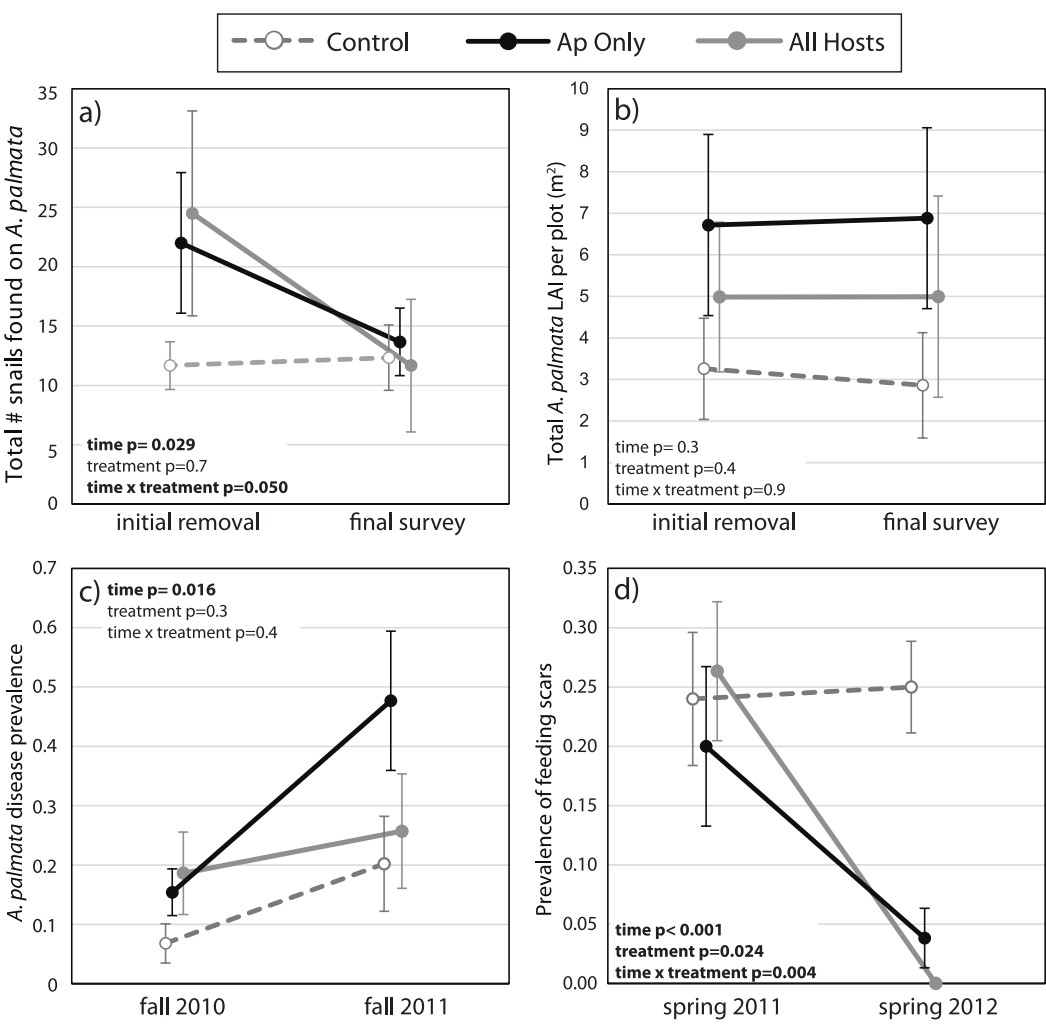

**Figure 2 Before-After-Control-Impact analysis.** (A) The total number of *Coralliophila abbreviata* found on *Acropora palmata* in a study plot, (B) the *A. palmata* tissue abundance as measured by the live area index (LAI, see text), (C) the prevalence of white disease on a random subset of *A. palmata* colonies during the seasonal peak in disease before and after the initial removal, and (D) the prevalence of *C. abbreviata* feeding scars on this random subset of *A. palmata* colonies at the survey prior to the initial removal and one year later. All points are mean ± standard error. Data were rank transformed for analysis and the *p*-values based on the transformed data are shown.

(± SD) years. Effect duration did not differ significantly between Ap Only removal (3.4 ± 1.4 years; mean ± SD) and the All Hosts removal (4.0 ± 1.4 years; mean ± SD) treatments (Wilcoxon, $Z = 0.31$, $n = 6$, $p = 0.8$).

## Size frequency/Sex ratios

A two-way ANOVA on log-transformed shell lengths (size) was used to compare the recolonized versus initial ApSnail populations between the two removal treatments (Fig. 5). Mean shell length was significantly larger at the initial (24.4 mm ± 7.8 mm, pooled mean ± SD; $F_{1,465} = 8.61$, $p = 0.003$; Figs. 5A and 5B) compared to the final (22.1 mm ± 5.7 mm, pooled mean ± SD; Figs. 5C–5D) surveys but not between the two removal

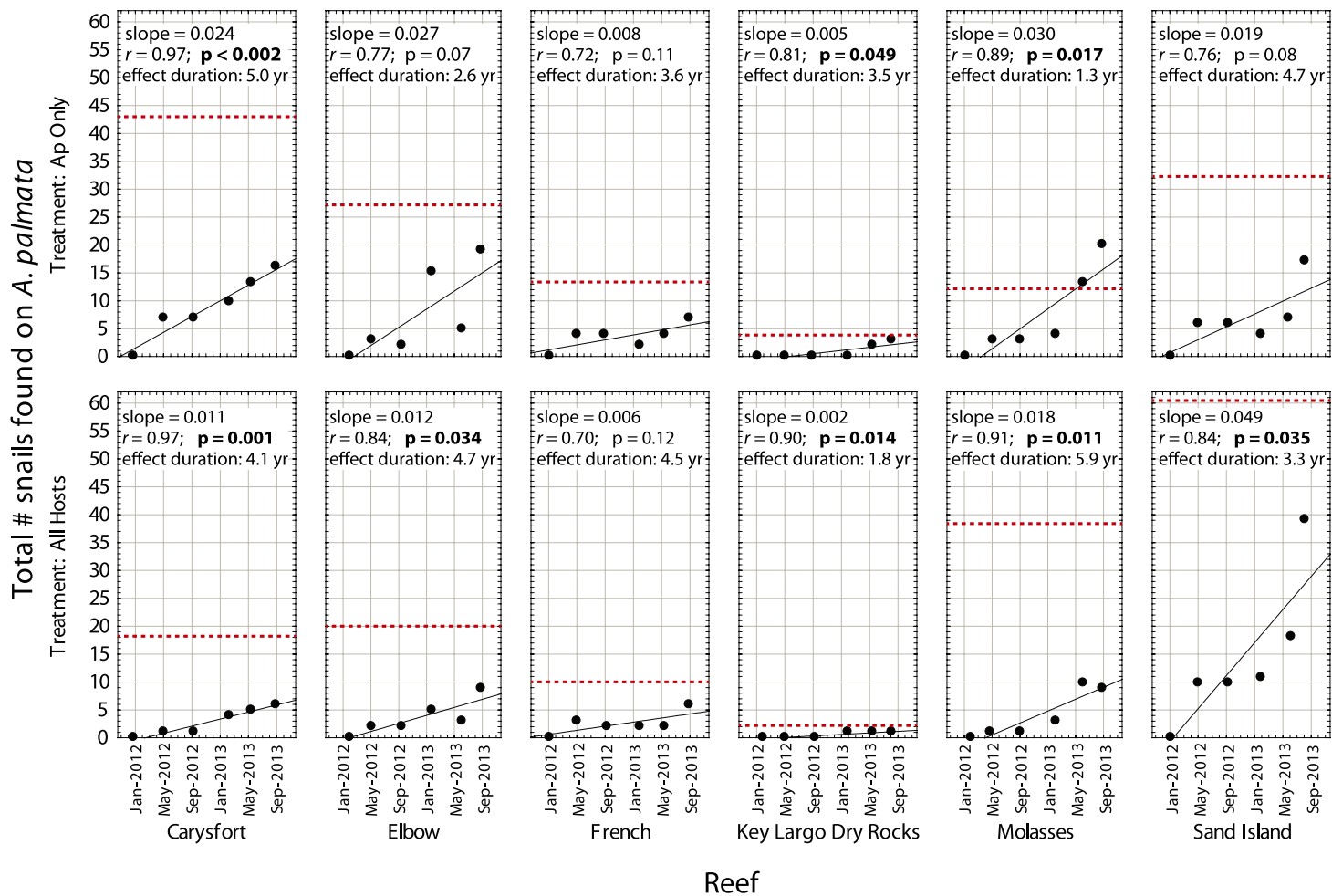

**Figure 3 Recolonization of *A. palmata* by *Coralliophila abbreviata* snails.** Number of *Coralliophila abbreviata* snails found on *Acropora palmata* in each plot where they were removed from *Acropora palmata* only (Ap Only) and from all host coral species (All Hosts) following the removal phase of the experiment. The dotted line indicates the baseline number of *C. abbreviata* that were removed from that treatment plot at the start of the experiment in June 2011. 'Effect duration' is the estimated time for the number of snails to reach the baseline (recolonization), according to the regression for each plot.

treatments ($F_{1,465} = 1.09$, $p = 0.3$) and there was no significant interaction between time and treatment factors ($F_{1,465} = 0.16$, $p = 0.7$). A separate two-way ANOVA was also used to compare the initial and final log-transformed shell lengths between ApSnails (pooled removal treatments) and the snails collected from other host corals in the All Hosts treatments (Fig. 6). Snails collected from other host corals were significantly smaller (18.8 mm ± 3.6 mm, mean ± SD; $F_{1,710} = 103.58$, $p < 0.001$) than those collected from *A. palmata* (24.4 mm ± 7.8 mm, pooled mean ± SD), and the snails collected at the initial survey were significantly larger (22.5 mm ± 7.2 mm, pooled mean ± SD; $F_{1,710} = 16.83$, $p < 0.001$) than those collected at the final survey (20.3 mm ± 5.9 mm, pooled mean ± SD), but there was no significant interaction ($F_{1,710} = 0.23$, $p = 0.6$) between host and time factors.

At the final survey, the proportion of males among ApSnails in the Ap Only treatment was 0.66 (54 males and 28 females) and 0.74 (52 males and 18 females) in the All Hosts

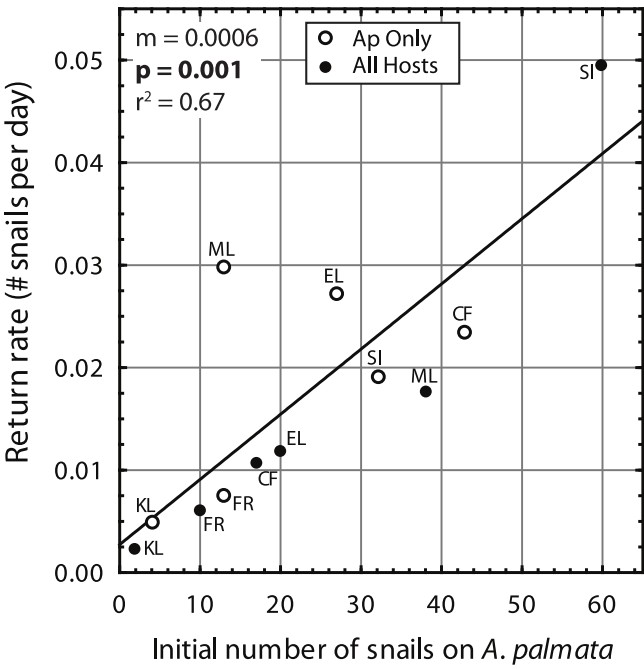

**Figure 4** *Corralliophila abbreviata* **recolonization rate.** Recolonization rate (number of snails per day) based on the slope of the linear regressions (Fig. 3) versus the number of snails found on *Acropora palmata* in each study plot at the initial removal.

treatment. Unfortunately, we do not have gender ratios for the initial population, but as larger snails are known to be female in this protandrous species (*Johnston & Miller, 2007*), the larger snail sizes of the initial population would be expected to reflect a lesser proportion of males relative to the final population.

## DISCUSSION

Corallivorous snail removal was effective in significantly decreasing both snail abundance and the prevalence of feeding scars observed 19 months following the removal. Consequent declines in disease prevalence and a parallel enhancement of total tissue abundance (LAI) might be expected, but both of these factors are strongly influenced by a multitude of additional known and unknown factors, thereby decreasing the ability to detect these parallel changes in the present study. However, based on the significantly lower prevalence of feeding scars we can deduce that less tissue was consumed by snails (*Miller, 2001*). Further evaluation of the effects of snail removals should focus on the performance of the *A. palmata* stand over a longer duration than was possible in this study.

The recolonization rate calculations project full recolonization to the baseline abundance of snails over nearly a 4-year period. High site-specific variability was observed in baseline abundance and recolonization rate. In some plots, the rate of snail arrival on *A. palmata* colonies may accelerate over time (Fig. 3), which is consistent with the aggregating behavior of snails (DE Williams, pers. obs., 2006). Thus the extrapolated 'effect duration' is an estimate and might differ in reality due to stochastic events or a non-linear response not observed during the duration of the experiment.

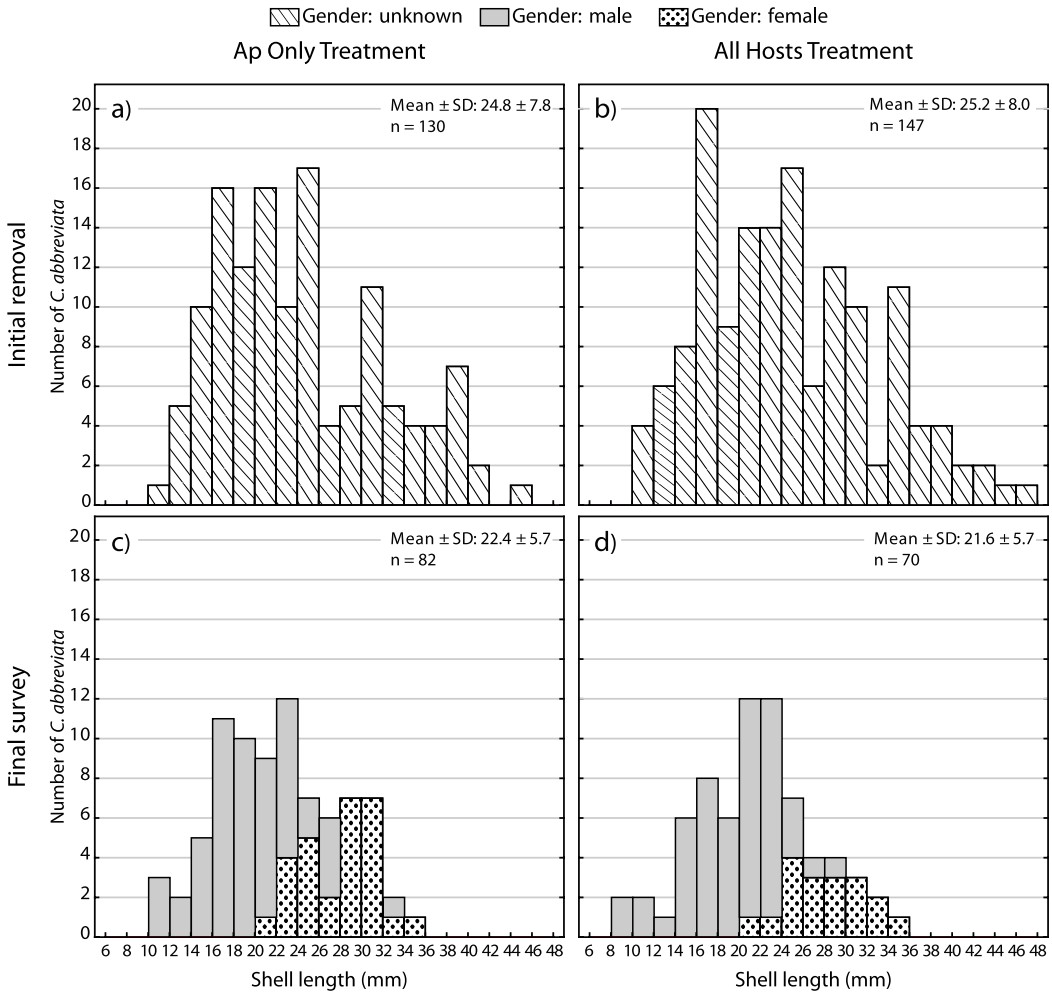

**Figure 5** *Coralliophila abbreviata* **size and gender frequency distribution.** Size and gender (Sept 2013 only) frequency distribution for the *Coralliophila abbreviata* snails collected from *Acropora palmata* host colonies in the (A) Ap Only (snails removed from *A. palmata* only) and (B) All Hosts (snails removed from all host coral species) treatments at the initial removal (June 2011) and at the final survey (Sept 2013) for (C) Ap Only and d) All Hosts treatments.

At the final survey, the average size of recolonizing snails was smaller than at the initial removal. Though the decrease in mean snail size was modest, the larger sized ApSnails (≥35 mm) were not observed to recolonize at all (Fig. 5) over the 19-month study period. At the initial removal, the maximum snail size was 47.0 mm and at the final survey the maximum was 34.6 mm. Tissue consumption by smaller snails is expected to be less than for larger snails (*Baums, Miller & Szmant, 2003b*; *Bruckner, 2000*; *Hayes, 1989*); thus, although this portion of the size distribution represented approximately 10% of the population, these larger snails likely were inflicting greater than 10% of the tissue loss associated with snail predation.

In addition to the direct impact of feeding, fecundity is disproportionately higher in these larger snails (*Johnston & Miller, 2007*). At the initial removal, there were 42 ApSnails found with a shell length >34.6 mm. Based on the relationship between shell

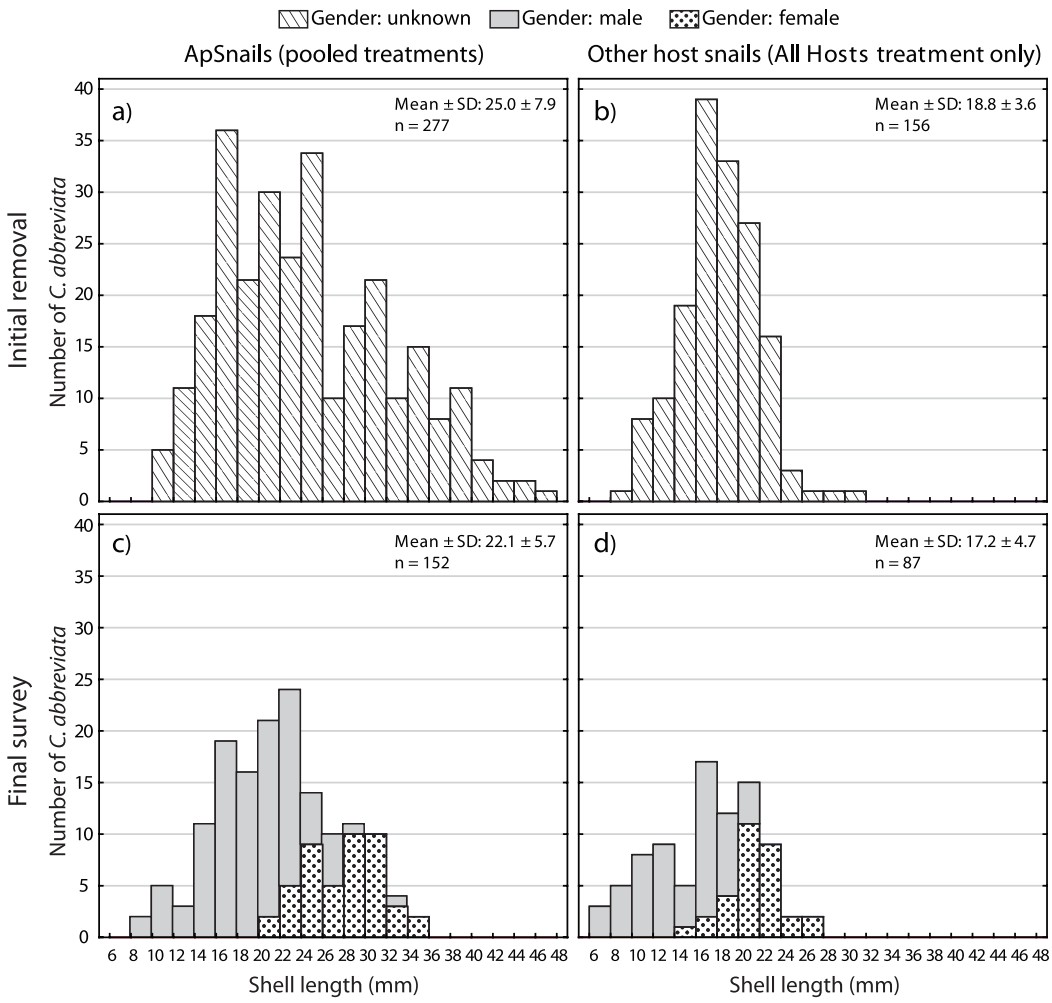

**Figure 6 *Coralliophila abbreviata* size and gender frequency distribution.** Size and gender (Sept 2013 only) frequency distribution for the *Coralliophila abbreviata* snails collected from (A) *Acropora palmata* host colonies and (B) other host species at the initial removal (June 2011) and at the final survey (Sept 2013) from (C) *A. palmata* and (D) other host coral species.

length and veliger production (*Johnston & Miller, 2007*), we can estimate that these 42 snails, assuming they were all females, would have produced >715,000 veligers in one reproductive cycle (clutch). For comparison, if we look at the largest 10% of the final size distribution (females ranging from 29.8 to 34.6 mm), these 17 females are expected to yield approximately 155,000 veligers in one reproductive cycle. In fact, all 46 female ApSnails found at the final survey combined would be predicted to yield 280,000 veligers in one clutch. Thus, the combination of fewer snails and the shift to smaller sizes could potentially decrease snail reproductive output by more than 50%. In the 3.7 years projected for full recolonization, it is possible that the snail size distribution will also shift back to the baseline size. Based on the size–age relationship reported by *Johnston & Miller (2007)*, the largest individual observed at the start of the experiment (i.e., 47 mm) might be approximately 15 years old while the maximum size observed at the final survey would be expected to be approximately 9 years old.

Individuals recolonizing *A. palmata* colonies appear to be primarily from surrounding reef substrate rather than larval recruits. The smallest recolonizing individuals found on *A. palmata* were 8 mm, and are expected to be ∼3 years old (*Johnston & Miller, 2007*), which is significantly older than the 1.5 years duration of the recolonization phase in this experiment (January 2012–August 2013). This suggests that these smallest snails may have been present but not detected at the time of the removal, hence were not new larval recruits. If other host corals were the predominant source of snails that recolonized *A. palmata* colonies, then we would expect a difference in size distribution and numbers of recolonizing snails between the two removal treatments (Figs. 5C and 5D), which was not the case. Although the snails on other host corals in the Ap Only treatment were not measured at the start of the experiment, they would likely have had the same size distribution as those collected in the All Hosts treatment at the start (Fig. 6B). Presumably, if these other host coral species were the source of recolonizing ApSnails, then the size distribution of ApSnails in the Ap Only treatment at the final survey (Fig. 5C) would be similar to the distribution of the snails collected from other host corals at the initial removal (Fig. 6B). Specifically, there were relatively few snails larger than 24 mm on the other host corals, yet more than one-third of recolonizing snails were larger than 24 mm (Fig. 6C). Instead, the size distribution of recolonizing snails in both removal treatments looks the same (Figs. 5C and 5D), suggesting that the snails found on *A. palmata* are not migrants from other host corals.

During the removal phase, the average number of snails found at the subsequent survey did not diminish (Fig. 1; mean of ∼2–5 snails), therefore it seems that the pool of colonizing snails was not depletable by reasonable removal efforts at these temporal and spatial scales. This effort was intended to reflect what could reasonably be achieved in a conservation application, thus only coral colonies of known snail-hosting species were systematically searched. Although the colony searches were thorough, snails on adjacent reef substrates between colonies would not have been detected until they recolonized an *A. palmata* colony at a subsequent survey.

In our experiment, no significant added benefit was derived from the additional effort required to remove snails from all host corals versus *A. palmata* only (Fig. 1). Removing snails from *A. palmata* only required 30 diver minutes versus 51 diver minutes for removal from all hosts. The density of other host colonies in the plots was $5 \pm 4.9$ colonies (mean $\pm$ SD) and a total of $21 \pm 15.6$ (mean $\pm$ SD) snails per plot were found on these other host colonies. If the site with the unusually high number of snails was excluded, the recolonization rates were marginally faster in plots where the snails were not removed from other hosts. It is possible that in areas where the other hosts species are more abundant or are harboring greater numbers of snails, the additional effort to remove them would be worthwhile, though more costly. However, the targeted removal of snails from only *Acropora* spp. hosts yields clear benefits yet allays concerns about potential unintended consequences of removal such as selection for earlier age of sex change, or potential trophic impacts on populations preying on the snails.

In planning snail removal as an *A. palmata* conservation effort, the cost (diver time) and benefit (reduced snail load) must be balanced. With the mean effect duration of nearly four years, one strategy could be to perform a removal at four year intervals. However, our results show high between-site variability in snail abundance; thus the strong correlation of recolonization rate with initial snail load (Fig. 4) suggests that the frequency or need for subsequent removals may be indicated by the number of snails that are found at the initial removal. In another view, the mean rate of snail recolonization appears to increase after one year (Fig. 1), so a single annual removal might be a useful target, at least for areas of high snail abundance (>0.2 snails m$^{-2}$ of reef area). Corallivorous snails feed more actively in warmer months (*Al-Horani, Hamdi & Al-Rousan, 2011*), so snail removal efforts could be more efficient during the summer. Additionally, although their egg production cycle is not well established, it is more common to find egg cases with mature veligers in mid-to-late summer (DE Williams, pers. obs., 2002); thus, removal prior to that may reduce larval production.

Most of our sites were located on spur and groove formations on the shallow fore reef. However, our study plots did not occupy the full extent of a reef 'spur', leaving contiguous reef areas populated with snails. In practice, removal from all corals on a contiguous spur may further prolong the effect duration of the removal. This may not be practical in areas where there are not natural breaks in reef structure, but removal from contiguous stands of *A. palmata* may be possible. The effect of *A. palmata* colony density on snail recolonization rate was not tested in this study; however, other studies have found that *C. abbreviata* abundance is generally lower in higher density 'thicket'-type stands of *A. palmata* (*Baums, Miller & Szmant, 2003a*; *Bruckner, Bruckner & Williams, 1997*; *Miller et al., 2002*). Therefore, removal effort could be more efficiently focused on *A. palmata* stands with intermediate or low colony density rather than dense thickets. Although removing snails is not technically difficult, *C. abbreviata* is fairly well camouflaged and divers should be trained to recognize them to ensure effective removal and to minimize collection of other non-corallivorous species such as *Thais deltoidea* that are commonly found around *A. palmata*.

Given the ecologically and legally imperiled status of *A. palmata*, and the intractability of managing the most severe threats such as disease and climate change, proactive conservation measures that can be implemented at a local level are needed. This experiment demonstrates the effectiveness of snail removal at a local scale with a 30 min diver investment to reduce corallivore loads over an estimated four year time scale in seven meter radius plots containing *A. palmata*. Given the rarity of *A. palmata* throughout many sections of its range, it is likely that two divers could successfully search for and remove snails over a large area of vulnerable *A. palmata* (i.e., low density stands with high snail loads) with only a few days of effort. While a wide range of conservation and restoration actions are required to address both local and global threats to corals, the removal of corallivorous snails may be used on natural or restored (outplanted) populations to conserve living acroporid coral.

## ACKNOWLEDGEMENTS

Field assistance by K Lohr, C Kiel and L Richter is greatly appreciated.

### Funding

This project was made possible by support from the NOAA Coral Reef Conservation Program and NMFS Southeast Regional Office, Protected Resources Division. The funders had no role in study design, data collection and analysis, decision to publish, or preparation of the manuscript.

### Grant Disclosures

The following grant information was disclosed by the authors:
NOAA Coral Reef Conservation Program.
NMFS Southeast Regional Office, Protected Resources Division.

### Competing Interests

The authors declare there are no competing interests.

### Author Contributions

- Dana E. Williams conceived and designed the experiments, performed the experiments, analyzed the data, wrote the paper, prepared figures and/or tables, reviewed drafts of the paper.
- Margaret W. Miller conceived and designed the experiments, wrote the paper, reviewed drafts of the paper.
- Allan J. Bright and Caitlin M. Cameron performed the experiments, reviewed drafts of the paper.

### Field Study Permissions

The following information was supplied relating to field study approvals (i.e., approving body and any reference numbers):

Florida Keys National Marine Sanctuary permit # FKNMS-2010-033, FKNMS-2010-130, FKNMS-2012-030.

### Supplemental Information

Supplemental information for this article can be found online at http://dx.doi.org/10.7717/peerj.680#supplemental-information.

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
