# Peer review of "Removal of corallivorous snails as a proactive tool for the conservation of acroporid corals"

_PeerJ, doi:10.7717/peerj.680_

## Round 0.1 · original submission · Minor Revisions

We now have 3 reviews from folks both familiar with the system and those from the Pacific who are coral reef conservationists but likely not as familiar with the system itself. All 3 referees responded and agreed to review the manuscript, and all 3 agree that it is a well-done and well-ritten study. There are a number of suggestions for improvement, and in particular, referee 3 has some questions regarding the experimental design, error quantification among observers, and diver effort .for effective restoration. I believe that these are valid questions, and ones other readers will likely have as well. Regardless, I expect that the authors should be able to deal with these questions rather simply during revision, and the uniform support of the referees for acceptance of a suitably revised manuscript is clear. I will look forward to seeing the revised manuscript...

·

Basic reporting

Line 8-The first paragraph needs some work to ease the reader in a bit better. Start with a sentence or two to provide the bigger picture for this work. Why is this work relevant?
Line 22- need an 'a' between provide and benefit
Line 32-needs a reference
Line 36-Include a referenced sentence indicating that these fishes have been overfished, because here the reader is making the assumption that this is the case for these species.
Also to link to the previous sentence better perhaps write “Therefore due to overfishing it is possible that the reduced predation on snails may have increased their abundance”
Line 38- Is the specific data missing to support this hypothesis due to lack of data on the overfishing or the snails?
Line 57-very interesting! Is it known at what age C. abbreviata starts producing offspring?
Line 58-add the acronym ESA after “Endangered Species Act” and spell out the acronym NMFS
Line 66- rather than writing a / just make it a list “ecological, legal and management conditions”

Experimental design

Line 76- Here briefly describe the topography of the six sites. Also there are no geographical coordinates (latitudes and longitudes) for each reef, which makes the work unreproducible, either put them in the text or Table 1 and refer to it (if you cannot include the site locations explain why).
Line 86- Please describe how you selected your plots within sites, as you control plots had a far lower initial # snails compared to the experimental plots (Figure 2).
Line 89-No need to include “(described in detail below)”
Line 90- Please describe why you decided to do 3 removals? It looks as though the initial one got most of the snails. In the discussion you suggested doing removals every 1-4 years, would practitioners have to do 3 removals each time?
Line 110- How was the length, width and height measured?
Line 125- Consider moving the permit information to the acknowledgements

Validity of the findings

Line 193- explain what “effect durations” are in the methods, why you use this metric and how they’re calculated.
Line 251- change “will” to “may” as this might be saying a little too much, because if individuals have reached a larger sizes before they can most likely do it again, especially if their natural predators have been fished out.
Line 252-When collecting the snails how did you look for them? Do the snails ever reside/hide underneath the colony? Also have you ever dived at night to see if more come out? Maybe the smaller ones avoid predation (and inadvertently divers) this way?
Line 291- by removal do you mean one dive (removal) per plot or 3 dives (as in your study).

Additional comments

A well written and executed piece of work with interesting and applicable results.

Reviewer 2 ·

Basic reporting

The basic structure of the article is fine. The writing is clear, the figures are appropriate, and the methods are understandable.

Experimental design

The authors present novel research on the question of how to reduce predation pressure of the corallivore C. abbreviata on threatened elkhorn corals. A BACI design is employed which is appropriate for the question. The sample size and survey methods are adequate and well established based on several prior studies conducted by Caribbean researchers. The BACI design is effective in controlling for high variability among reefs in coral and snail abundance.

Validity of the findings

The study provides robust support for removing snails from coral colonies to reduce snail numbers on A. palmata. This is important because many of the other stressors are not easily manageable (such as increasing disease prevalence). However, I have several questions about the interpretation.

1) The control plots from the start had a much lower density of snails. Why was that? Where they chosen that way or was that chance? When looking at Figure 1 there are two explanations to the casual observer for why the control plots have lower snails then the removal plots. Either they had lower numbers of snails by chance or visual surveys of snails discover fewer snails then when divers actually remove them. I think you said you measured all snails even the ones from the control plot. That would exclude the second explanation. I would add a sentence along those lines to the discussion just to emphasize that it wasn’t the survey method.
2) Were they coral densities comparable among the plots within sites? Between sites? Host density is known to affect snail density as the authors mention in the discussion and might have to be added as a co-variate in the analysis.
3) In the discussion I think there needs to be a paragraph on long-term effects of such removals. For one, the effectiveness of snail removal could not be shown in this study to help preserve coral tissue although this is likely to be the case in the long run. Thus, in any future removal study, this aspect needs to be monitored. Second, it is well known that targeting large individuals (females) in protandrous species (i.e. shrimp, some fish) leads to selection for earlier sex change. This is not likely to be the case in the short term but such an effect needs to be discussed as a potential (unwanted?) outcome and monitored (“size at sex change”) in any future removal studies.

Figure 3 where is the dotted line in the Sand island all host plot?

Line 269 this needs a conclusion sentence: thus the recolonizing snails likely came from other Ap colonies outside the 7 m plots?

Line 279 Or the previous search was insufficient to remove all snails
Line 314 You need to add a sentence here saying that re-evaluation of the effectiveness of snail removal in reducing tissue loss should be conducted at regular intervals – large scale removal of preferentially large snails is expected to exert selection patterns on protandrous hermaphrodites to make the reproduce earlier.

Additional comments

This is a well written and conducted study on an interesting management question.

·

Basic reporting

The article adheres to PeerJ policies. It is written in a clear and concise format that is easily understood.

Experimental design

The overall experimental design is solid with a few exceptions listed below. Taking into consideration the seasonal and annual temporal variation in disease and feeding scar levels reduces overall error. Site selection was good enough to provide a wide range of results and plots included both clumped and widely distributed colonies to better assess effort.

For the measure of partial mortality of corals the authors state that length, width, height, and, % live visual estimates (line 110) were recorded. There can be considerable variation between observers in determining percent live cover. Was observer error measured and calibrated between divers or did the same diver estimate percent live cover?

In lines 148-151 the authors assumed the number of snails following the final removal phase to be zero. Another diver going back to same plot in same dive to determine if any snails remained could have easily tested this after each of the three removals. The number of snails found on last removal was not zero and were attributed to recolonization rather than missed snails. In nearly half of the control sites where snails were counted but not removed there were less snails there on the final count. This is not overwhelming evidence for recolonization. Is this based on the smaller average size of snails in the final counts?

The authors state there was high variability between sites and colonies. The standard deviations and error bars in the graphs support this. Would a larger sample size have reduced some of this variability? Was a power analysis to determine the correct number of plots to be able to show a difference conducted prior to the surveys? The worst case scenario in variability should drive the number of samples.

Validity of the findings

The authors mention in lines 28 and 29 that disease, storms, and bleaching have been the major cause of decline to the reefs in the western Atlantic. Bleaching events are predicted to be more frequent and severe in the near future. The impact of snail predation is not in the same threat level as these global events.
In a natural system, snails are not a real threat but with such low populations of A. palmata the authors argue this may be a viable method for coral recovery. The time and effort would be better spent combatting these larger threats. Many managers feel they are powerless against these threats. However there are a number of effective strategies that are being used to address the effects of climate change to coral reefs. Managers are identifying areas and species of resilience and creating protected areas to isolate them from anthropogenic impacts. There are currently strategies to reduce the amount of light and increase water circulation during bleaching events. There are also methods in place to reduce physical stress to corals such as limiting activities during these events. Recovery processes are also being explored. All of these efforts are being implemented in Florida (NOAA Responding to Climate Change). There is no comparison of costs and effort with other established methods of restoration in the manuscript.

The authors state in their abstract “ However, corallivory by C. abbreviata is one of the few major sources of partial mortality (contrasting with threats such as bleaching, disease, or storm disturbances) that can be locally managed”. There are many sources of partial mortality that can be locally managed. The authors do not mention other local threats that may or may not pertain to their region such as sedimentation, nutrification, coastal development, overuse etc. Restoration can include direct interventions such as transplantation or passive management techniques that remove impediments to natural recovery such as watershed management. In the Florida Keys NMS restoration projects address human induced impacts such as vessel groundings and coastal construction.

“‘Effect duration’ is the estimated time for the number of snails to reach the baseline (recolonization), according to the regression for each plot.” Fig. 3 extrapolates out to time it takes to go back to baseline and ranges from 1.8 to 6.9 yrs. The experiment time after removal is <2 yrs. How accurate is it to predict out to nearly 7 yrs. There could be a number of stochastic events that could disrupt this pattern. A short discussion on why this is valid would be helpful.

Additional comments

The manuscript is well written however it needs some revision. There is good detail for a spatial and temporal management plan for snail removal including useful size and fecundity background information. The main value in this research is taking it from a colony level assessment conducted previously to a community level that can be implemented by managers on a large scale making the results more applicable to successful management.

This article can be improved by including information about other restoration techniques and efforts being used in Florida and comparing the effort and success involved.

I do not see where the authors mention what the smallest size is that snails can be detected. The smaller sizes are likely very cryptic. From Fig. 5 it appears that 8 mm was the smallest collected which is approximately 3 years old.

One graph instead of both Figs 5 and 6 is all that is needed. I suggest removing Fig 6 and just explain in the text.

The authors should include at what spatial scale this management strategy would be conducted. If it takes 30 diver minutes for a 7 m radius that would be a huge time intensive effort over a large area. This does not appear to be the best use of restoration efforts.

Federal Register references out of alphabetical order

---

## Round 0.2 · accepted · Accept

Having read through your revised manuscript I am satisfied with the revisions and that you have taken the comments of the referees to heart. Overall, I think the edits address the primary criticisms of the referees sufficiently that I see no need to send the manuscript out for additional review when all suggested only minor revisions and were unanimously positive about the manuscript in the first round.
Having said that, there are some points of contention among the referees (such as whether or not to combine Figs 5 & 6) and between the referees and authors (such as whether or not to review other restoration techniques in the discussion or focus on just this proposed method of control). I can see both sides of these preferences, but in my opinion they are just that - personal preferences, rather than issues of scientific disagreement. As such I am inclined to side with the authors to include the content that they feel is most appropriate in their manuscript. Given that I see no remaining scientific issues following revision in response to referee comments, I am happy to accept your manuscript and move it along for publication.